

# Multiscale transport and 4D time-lapse imaging in precision-cut liver slices (PCLS)

Iqra Azam and James D. Benson

Department of Biology, University of Saskatchewan, Saskatoon, Saskatchewan, Canada

## ABSTRACT

**Background**. Monitoring cellular processes across different levels of complexity, from the cellular to the tissue scale, is important for understanding tissue structure and function. However, it is challenging to monitor and estimate these structural and dynamic interactions within three-dimensional (3D) tissue models.

**Objective**. The aim of this study was to design a method for imaging, tracking, and quantifying 3D changes in cell morphology (shape and size) within liver tissue, specifically a precision-cut liver slice (PCLS). A PCLS is a 3D model of the liver that allows the study of the structure and function of liver cells in their native microenvironment.

**Methods**. Here, we present a method for imaging liver tissue during anisosmotic exposure in a multispectral four-dimensional manner. Three metrics of tissue morphology were measured to quantify the effects of osmotic stress on liver tissue. We estimated the changes in the volume of whole precision cut liver slices, quantified the changes in nuclei position, and calculated the changes in volumetric responses of tissue-embedded cells.

**Results**. During equilibration with cell-membrane-permeating and non-permeating solutes, the whole tissue experiences shrinkage and expansion. As nuclei showed a change in position and directional displacement under osmotic stress, we demonstrate that nuclei could be used as a probe to measure local osmotic and mechanical stress. Moreover, we demonstrate that cells change their volume within tissue slices as a result of osmotic perturbation and that this change in volume is dependent on the position of the cell within the tissue and the duration of the exposure.

**Conclusion**. The results of this study have implications for a better understanding of multiscale transport, mechanobiology, and triggered biological responses within complex biological structures.

Corresponding author
James D. Benson,
james.benson@usask.ca

# INTRODUCTION

Cryopreservation of biological specimens could provide tissues and organs on demand for transplantation, toxicological, and pharmacological studies. However, current tissue cryopreservation techniques require high concentrations of toxic and mechanically

disrupting cryoprotectant agents (CPAs) (*de Graaf et al., 2007*; *Fahy et al., 2013*; *Best, 2015*; *Giwa et al., 2017*). These concentrated solutions cause a complex osmotic response including shrinkage and swelling of the tissue resulting in mechanical damage. Osmotic stress is known to affect structure and function and is associated with cell damage, disruption of cell–cell interactions, deformation of the extracellular matrix (ECM), and alteration of metabolic activity (*Corasanti, Gleeson & Boyer, 1990*). For instance, during hyposmotic stress, fluid flows into the cell, leading to cell swelling, membrane blebbing, and an increase in intracellular hydrostatic pressure (*Sardini et al., 2003*). Consequently, this swelling of a cell may result in changes in gene transcription through calcium-regulated mechanotransduction signaling pathways and relocalization of transcription factors due to the connection between the cytoskeleton and protein complexes that control the opening and closing of mechanosensitive ion channels (*Okada et al., 2001*; *Nilius, Watanabe & Vriens, 2003*). Conversely, hyperosmotic stress and cellular shrinkage can reduce intracellular hydrostatic pressure, leading to pathologies such as cytoplasmic stiffening (*Prystopiuk et al., 2018*; *Han et al., 2019*), DNA damage and apoptosis (*Burg, Ferraris & Dmitrieva, 2007*), oxidative stress, inflammation (*Vaziri & Rodríguez-Iturbe, 2006*), and cellular aging (*Myers et al., 2007*). However, the effects of osmotic stress in animal tissues and its biological consequences are relatively unknown.

One approach to improve tissue preservation methods is the use of mathematical modeling approaches to predict an optimal method for equilibration to and from high concentrations of CPA, also known as CPA loading and unloading (*Warner et al., 2021*). Modeling cryobiological protocols requires accurate estimation of the state of cells and nuclei as a function of time, including cell and nuclear volume and concentration. Moreover, successful damage modeling requires identifying the different cell types within a tissue as well as identifying subcellular and extracellular structures. However, it is challenging to estimate these parameters in complex 3D tissues due to the limited availability of non-invasive methods to measure the effects of mechanical stresses within a complex multicellular environment (*Warner et al., 2021*). Therefore, this study was designed to develop modern real-time imaging techniques and image analysis methods to quantify the effects of osmotic stress in tissues. Portions of this text were previously published as part of a thesis (https://harvest.usask.ca/server/api/core/bitstreams/659a09e4-fb3e-458e-a34d-b006ebd9d981/content).

Conventional histological techniques have been used to estimate the morphological or structural properties of tissues. These techniques require destructive tissue sectioning and result in unquantified shrinkage and swelling of tissues during histological processing. However, confocal laser scanning microscopy (CLSM) overcomes these limitations by offering non-invasive, high resolution 3D imaging of thick tissue sections (*Jones et al., 2005*; *Enyedi et al., 2013*). It facilitates the *in-situ* characterization of the tissue architecture by employing immunofluorescence (*via* a fluorophore) in three spatial (*e.g.*, *xyz*) dimensions. Using this microscope, optical sectioning enables the acquisition of a series of focused images along the objective's *z*-axis with fixed increment steps. Later these acquired *z*-stacks of images can be combined using 3D image reconstruction software. This allows the quantitative measurement of volume and 3D reconstruction of cell and
matrix microstructure (*Chvátal et al., 2007*). However, previous studies have found that homogeneous staining is still limited to a thickness of 100 μm in densely packed tissues such as the liver (*Johnson & Rabinovitch, 2012*). It is, therefore, necessary to section large volumes of tissue into serial precision cut slices that are stained and imaged separately to obtain high-resolution data throughout a tissue's thickness.

A 3D cell culture technology for studying cell behavior in its native microenvironment is *ex vivo* precision-cut tissue slices (PCTS) derived from various organs and species (*Burgstaller et al., 2015*). For physiology and toxicological studies, vital PCTS from human and animal organs, especially the liver, intestine, and lung, have been most widely used (*Hashemi et al., 2008*; *Liberati, Randle & Toth, 2010*; *Van Midwoud et al., 2010*; *Fisher & Vickers, 2013*). In addition to studying individual cells and cell layers, it is important to study how cells interact with their surrounding extracellular matrix (*Burgstaller et al., 2015*). The analysis of tissue compartments in living PCTS, as well as subcellular structures, requires the staining of cells or nuclei, followed by capturing 3-dimensional *z*-sections over time, a technique called four-dimensional (4D) imaging. However, it is still unclear how 4D time-lapse microscopy can be used to study the dynamic volumetric responses of the cells in a heterogeneous tissue.

To study cell volume changes in a live tissue, a live staining method must first be established for cell morphology visualization (*Johnson & Rabinovitch, 2012*). Fluorescent probes enable the study of the structural organization of the cells and subcellular structures as well as for functional measurements (*Johnson & Rabinovitch, 2012*). However, there are multiple challenges to visualize cellular level morphological parameters within a tissue, such as stain penetration, imaging depth and speed, fluorescence crosstalk, photobleaching, and phototoxicity. Moreover, it is challenging to maintain cell and tissue viability throughout imaging due to hypoxia-induced cell death. Hence, a major objective of this study is to adapt faster high-resolution imaging methods offered by modern confocal microscopy equipped with resonant scanning for high-speed real-time imaging.

The nucleus, traditionally considered a storehouse for genetic information, has recently emerged as a dynamic player in cellular responses. It acts as an active component of the cell, converting mechanical stresses into signaling output (*Lomakin et al., 2020*). Under mechanical forces, mechanical properties of the nuclei often dominate cellular behavior and can influence gene expression (*Fedorchak, Kaminski & Lammerding, 2014*). A recent study has shown that nuclei can be used as a probe for compressive forces and can be employed to determine the effects of external osmolarity on 3D cell-spheroids (*Khavari & Ehrlicher, 2019*). This "ideal" nuclear probe is defined as a uniformly scattered, endogenous sphere within a densely packed tissue space; however, this use of nuclei as a probe has not been investigated for thick (>100 μm) heterogeneous tissues.

Finally, cell- or agent-based models are a class of computational models that capture cell-level detail of multicellular tissues (*Macal & North, 2005*; *Wang et al., 2015*; *Soheilypour & Mofrad, 2018*). These models build tissues out of cell-agents that independently interact with their neighbours and with their extracellular and interstitial environment *via* rules that capture biomechanical physics and transport. This approach is powerful as it allows for specification of separate rule classes for different cell types, arrangement of cells

into anatomically accurate tissues, and tracking of cell states such as volume, apoptosis, and necrosis (*Macklin et al., 2012*; *Benson, Benson & Critser, 2014*; *Jenner et al., 2022*). A challenge with agent-based models is accurately informing model parameters: observations of cell behaviour *ex situ* does not always correspond to cell behaviour *in situ* (*Fry & Higgins, 2012*). Therefore, there is a need for non-invasive *in situ* observations of individual cell responses in tissues.

The present study couples high resolution imaging offered by confocal microscopy with a quantitative assessment of morphometric parameters in living liver tissue slices. Our primary objective was to design a method for imaging, tracking, and quantifying 3D changes in cell morphology (shape and size) within liver tissue, specifically a PCLS. A PCLS is a three-dimensional (3D) model of the liver that allows the study of the structure and function of liver cells in their native microenvironment. We combine techniques including fluorescent staining, confocal imaging, optical sectioning, automated segmentation, 3D reconstruction and surface rendering. All these techniques were used to investigate morphological properties during exposure to osmotic perturbations in liver slices. We used two approaches to quantify the effects of osmotic stress within tissues. First, we used endogenous cell nuclei as pressure sensors (probes) and confocal microscopy to estimate real-time nuclei deformation or change in position under osmotic stress. Secondly, we used microscopic imaging and image analysis techniques to estimate the changes in cell volume as a function of time and solute concentration. These data are critically important to informing multiphasic transport driven cell and tissue mechanics models such as one developed by *Warner et al. (2021)*. The outcome of this study has implications for a better understanding of multiscale transport, mechanobiology, and triggered biological responses within complex biological structures.

# MATERIALS & METHODS

## Animals

C57BL/6J mice (12–14 weeks old, weighing about 30–35 g) were used in this study and all animals were housed at the Lab Animal Services Unit (LASU) at Health Sciences Department, University of Saskatchewan, Canada. Animals were housed at 22 °C, with 60% relative humidity, and a 12-hour light/dark cycle. All animal use in this study was approved by the University of Saskatchewan's animal ethics committee under a tissue share agreement and exempt from animal use permits. Experiments conducted this study are in strict accordance with the guidelines provided by the Canadian Council on Animal Care.

## Preparation of precision cut liver slices

Adult anesthetized mice were sacrificed for other purposes by cervical dislocation prior to hepatectomy. Following that, the left liver lobe was excised and placed in a cold oxygenated excision buffer to prevent hypoxia. For preparation of liver slices, a vibratome (model VF-310-0Z; Precisionary Instruments, Ashland, MA, USA) was used to produce even, consistent slices through fully automated slicing (*Fisher & Vickers, 2013*; *Koch et al., 2014*). Before sectioning, liver tissue was transferred into a tube with a 2% solution of low melting temperature agarose (low EEO), followed by cooling on ice for 5 min, to allow gelling of the

agarose for holding the liver tissue. During sectioning, vibratome speed was set at two and the oscillation at four for the liver tissue. Precision cut liver slices (PCLS) of 300–500 μm thickness were prepared and immediately placed in a refrigerated oxygenated excision buffer following a method by *de Graaf et al. (2007)*. To keep liver tissue viable, efforts were made to remove the liver immediately after euthanasia and keep media aerated and cold to prevent hypoxia.

## Solutions

The excision buffer consisted of Hank's Balanced Salt Solution (1x HBSS) with 8.9 mM sodium bicarbonate, supplemented with 10% FBS and 5% penicillin streptomycin. Moreover, incubation media consisted of Eagle's Minimum Essential Medium (MEM) with 1.5 g/L sodium bicarbonate, non-essential amino acids, L-glutamine, and sodium pyruvate, supplemented with 10% FBS and 5% penicillin streptomycin (all media from Sigma-Aldrich, St. Louis, MO, USA). The solutions were continuously bubbled with air to keep them oxygenated (aerated) and maintained at a pH level of 7.4. Hyper- and hyposmotic solutions containing only non-permeating solutes were prepared by adding salt (NaCl) or distilled water to the isosmotic phosphate-buffered saline (PBS) to create solution osmolalities of 10, 80, 617, and 1,215 mOsm/kg. For permeating solutes, a solution was made by adding 1 mol/kg DMSO (ThermoFisher Scientific, Waltham, MA, USA) to isosmotic PBS. All osmolalities were measured using a Fiske 2400 multisample freezing point depression osmometer (Fiske Associates, Norwood, MA, USA).

## Staining

Cell membrane, cytoplasm and nuclear staining was accomplished using the CellMask plasma membrane stain, calcein-AM, and Hoechst dyes, respectively. CellMask contains a lipophilic moiety that binds excellently to membranes and a hydrophilic dye that anchors onto the membrane. Similarly, calcein-AM is a membrane permeable dye that is converted into the highly fluorescent membrane-impermeable calcein where its AM group is cleaved by intracellular esterases in the cytoplasm of live metabolizing cells. Further, Hoechst 33342 nucleic acid stain emits a blue fluorescence when bound to dsDNA and is a popular cell-permeant nuclear stain. We selected CellMask (red), calcein-AM (green), and Hoechst 33342 (blue) dyes to avoid overlap between their emission spectra.

A staining buffer was prepared by adding Deep Red Plasma Membrane Stain CellMask (5 μM, catalogue number: C10046), calcein-AM (5 μM, catalog number: C1430), and hoechst33342 (5 μg/ml, catalog number: H3570) (Thermo Fisher Scientific, Waltham, MA, USA) solution in an incubation media. After sectioning, tissue slices were incubated in a 6-well plate half-filled with cold staining buffer on an orbital shaker for 10 min. Finally, tissues were washed twice in fresh EMEM media supplemented with 10% FBS.

## Image acquisition

*In situ* imaging was performed with either the Nikon AXR confocal laser scanning microscope (CLSM) or the Zeiss 880 Airyscan microscope for high-speed imaging equipped with an Ar/HeNe laser. The Nikon AXR was used with a 4 × objective to capture the whole tissue at once. For better detail, the 20 × objectives of both microscopes were used to

achieve improved resolution and cellular level magnification. Slices were mounted on the stage of the confocal microscope in chambered cover glass slides (Thermo Fisher Scientific, Waltham, MA, USA) for fluorescence measurements and imaged using standard filter sets and laser lines. To hold tissue slices during imaging and dispensing liquid, a hydrophobic pen was used to make a well, as shown in Fig. 1A, and a small six mm microcoverglass (Arther H Thomas company, Philadelphia, PA, USA) was placed on top of the liver slice. The slices were kept at 4 °C during and before measurement throughout the experiment. All images were captured at 514 × 514-pixel resolution at 10–15 µm $z$-direction intervals at excitation/emission maxima at 649/666 nm, 488/525 nm, 350/460 nm for CellMask, calcein-AM, and Hoechst 33342 respectively. The number of images in the $z$-stack varied based on the thickness of the liver slices, which ranged between 300 to 500 µm. To accommodate bumps and curves in the tissue, scanning was extended to around 10 µm beyond the expected tissue thickness. Prior to the observation, cells or tissue were kept in EMEM medium containing 10% FBS.

For experiments where tissues were rapidly exposed to anisosmotic solutions, culture media was carefully replaced with different permeating and nonpermeating solutions within the channel using a Pasteur pipette for liquid dispensing without physically disrupting the tissue being imaged. The solution changes occurred within 2–3 s. In this study, Hoechst 33342 marked volume was considered to be the nuclear volume, and the whole cell volume is taken to be that of the CellMask marked volume with calcein-AM as a marker of tissue or cell viability (Fig. 1B).

## Data and image analysis

Data sets for confocal 4D projection were either projected with ZEN 2.3 (blue edition, Carl Zeiss, Baden-Württemberg, Germany) or Nikon NIS elements software (Nikon Instruments, Tokyo, Japan) or imported to Imaris 9.0 and 9.8 software (Bitplane, Oxford Instruments, MA, USA). Images were compiled using these software packages to save each $z$-stack file as a tiff file for all time points. For individual cell segmentation within a tissue, the 3D multi-$z$ TIF files exported from NIS and Zen software were segmented with Cellpose 1.0 using the cyto (cytoplasm) and cyto2 models with both channels (*Stringer et al., 2020*). When the cyto model was selected, the nuclear channel was chosen as an additional option to guide segmentation. The segmented images were saved both as outlines and as masks using Cellpose 3D (*Pachitariu & Stringer, 2022*). With Cellpose-Fiji (ImageJ), cell segmentation was performed with Cellpose, and quantification was done with Fiji. Following segmentation, additional size filtering was performed with morphoLibJ to remove objects containing less than 10,000 voxels from the segmented saved masks. To render the segmentation and convert them into ROIs (regions of interest), the labeled images were colored using the *glasbey_inverted* LUT. After analyzing the ROIs, MorphoLibJ was used to render the volume measurements from the 3D regions. For rendering time lapse movies, Fiji 3D viewer and Imaris were used.

Furthermore, for 3D reconstruction and volume rendering of whole tissue, the surfaces method in Imaris 9.0 was used. The surface was created using a threshold of absolute intensity, and radii were calculated by measuring half the longest diameters of the tissue

**A**

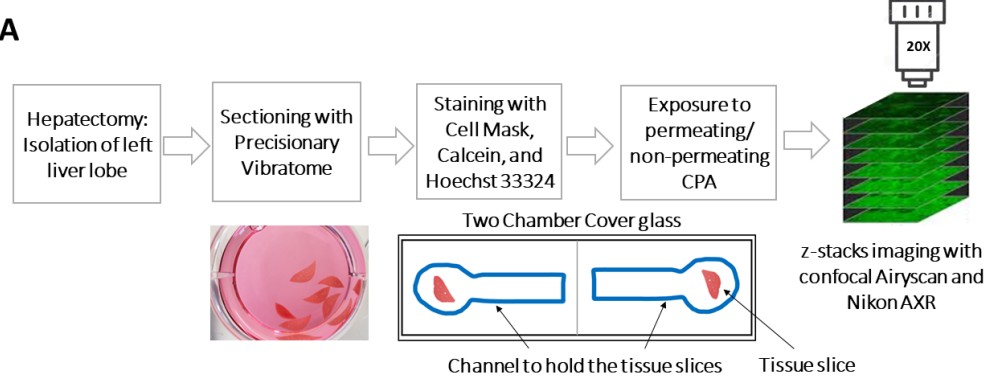

**B**

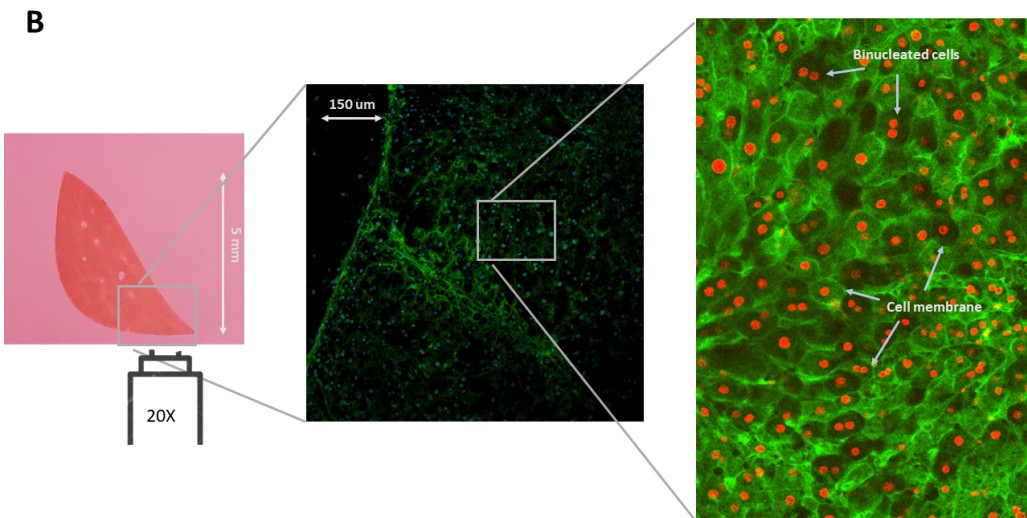

**Figure 1** **An illustration of the experimental procedure for imaging liver tissue.** (A) The process of acquiring liver tissue, followed by the preparation of PCLS, staining of the liver tissue, and the acquisition of 3D images. The blue well represents the line drawn by hand with a hydrophobic pen to hold tissue slices during imaging and dispensing liquid. (B) A two-dimensional representation of the size of the precision cut liver slice as well as the staining of the membrane and nuclei of the cells with CellMask (green) and Hoechst 33342 (blue; projected as red by Cellpose). A magnified and labelled image also shows the binucleated cells within the liver tissue. Note: The diffuse green color in the image represents cell membrane staining, while the projected red color, which comprise mostly solid disks, indicates nuclei staining. This distinction is also highlighted in the labeled image of the sample.

using SliceView in Imaris. Imaris software rendered the confocal 4D data sets as volume or surface and exported the data.

Finally, nuclei tracking was done by using an Imaris automated tracking method, *i.e.,* the spot method shown by the Hoechst 33324 staining. Time series three-dimensional files were directly analyzed with Imaris for nuclear volume analysis and for tracking nuclear movement throughout time. This analysis outputs nuclear position, volume, velocity, and acceleration within a three-dimensional tissue. To analyze and visualize the data obtained from the image analysis, R-packages (RStudio) were used for data visualization and statistical analysis.

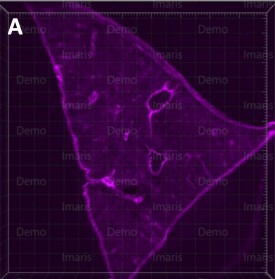 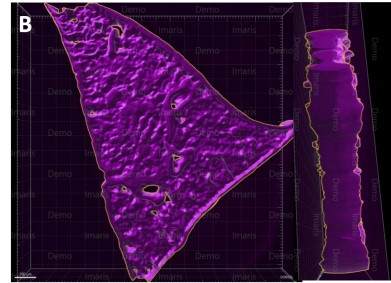 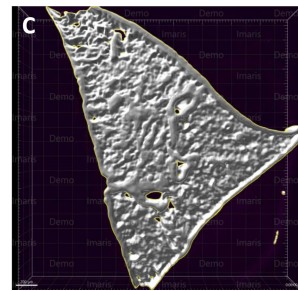

**Figure 2** **A microscopy image of mouse liver lobe.** An image of the left liver lobe of a mouse liver (300 µm thick), sectioned by vibratome for the purpose of imaging (A). Using the Imaris surface method, we rendered the volume data for liver slices, in which panel (B) shows the surface outlined by Imaris as a yellow-marked outline. The whole surface was shown with a white masked surface (C), demonstrating the surface masked tissue, which is used to render the area and volume of the tissue.

## RESULTS

### Quantification of volume changes at whole tissue level

Taking advantage of the Nikon AXR's wide field of view and faster speed, we acquired in-depth $z$-sections of full left lobe slices, at a $z$-step interval of 10–15 µm. To quantify changes in tissue volume during equilibration with cryoprotectant and anisosmotic media, tissue slices were allowed to equilibrate for 15 min with both media. Prior to and after equilibration, $z$-stack images of whole tissue were acquired that were later reconstructed to render volume information by using a surface method in Imaris (Fig. 2). Moreover, as shown in (Fig. 3), when we plotted the volumetric responses of the liver tissue, we observed that tissue volume decreased substantially after the addition of 10, 30, and 50% DMSO compared to its initial isosmotic media. Since mouse liver tissue contains vasculature and sinusoids lumen, the initial volume of each slice differs slightly. In the presence of hyperosmotic non-permeating solutes, we observed a significant decrease in tissue volume. In contrast, treatment with hyposmotic non-permeating media resulted in a slight increase in tissue volume, suggesting that whole tissue undergoes expansion or shrinkage when exposed to osmotic perturbations.

### Quantification of changes in nuclei position within a liver tissue

We tracked nuclei position in 3D tissue slices as represented in Fig. 4, and we observed a significant change in the position of nuclei upon exposure to anisosmotic and cryoprotectant media. For instance, nuclei tracking in hyposmotic media shows whole-tissue swelling and in hyperosmotic media shows whole-tissue shrinkage (Fig. 5). Our results suggest that nuclei are subjected to directional movement upon addition of anisosmotic media. When tissue was exposed to hyposmotic media, nuclei moved upward as a result of the low concentration of solutes at the top surface (Fig. 5A). Conversely, when tissue was exposed to permeating media of high osmolarity, nuclei move downward (Fig. 5B).

To compare different regions of tissue, we spatially segmented the tissue into two groups, periphery, and core. When tissues are exposed to anisosmotic media, diffusion occurs at the periphery of tissue, resulting in greater shrinkage at the edges than at the core. We

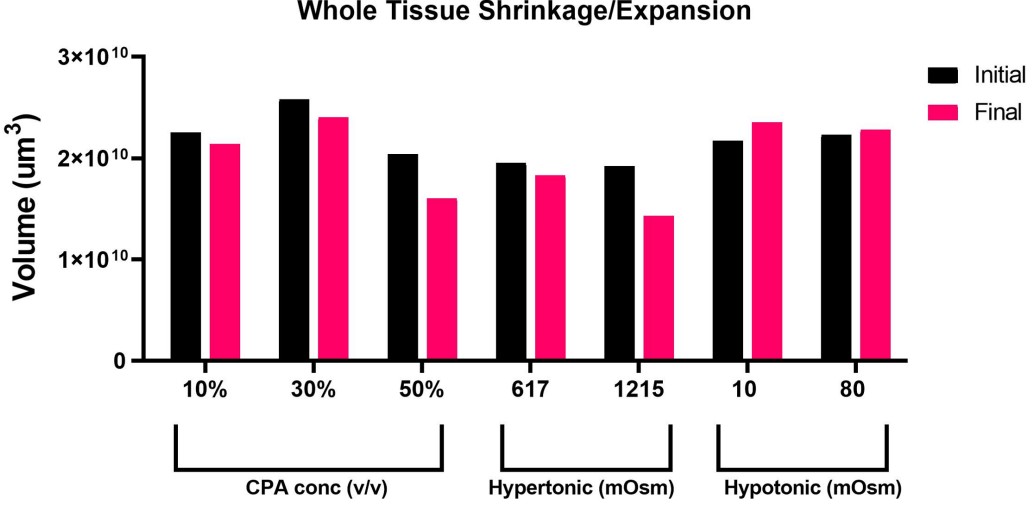

**Figure 3 Volumetric responses of whole tissue slices.** A comparison of volumetric responses of whole tissue slices ($n = 1$ each treatment) from initial to final time, quantified through $z$-stack imaging of thick ($\sim$300 μm) tissue sections, followed by 3D reconstruction and volume rendering.

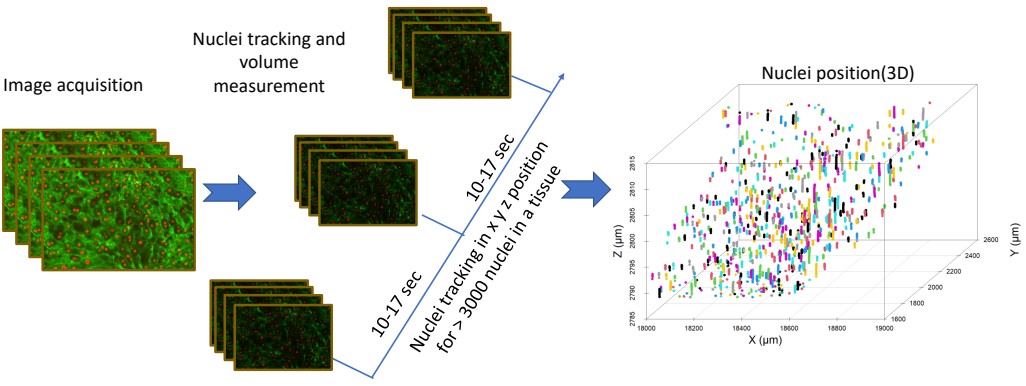

**Figure 4 Image analysis flow diagram.** The steps involved in acquiring nuclei tracking data from liver tissue can be seen in this flowchart. Nuclei tracking was done in four dimensions ($x, y, z$, and time), different colors represent the tracking IDs for the nuclei ($n = 800$).

illustrate this in Fig. 6A where nuclei paths are colored according to their positions and change over time, which represents more shrinkage at the edge of the tissue as indicated by the red colored tracking tails. Furthermore, in the core of the tissue, Fig. 6B shows that there is nuclear movement in different directions, likely caused by the presence of complex vasculature, sinusoidal, and bile ducts, which results in differences in osmotic pressure.

## Quantification of individual cell volume changes in liver tissue

We measured cell volume change with respect to its spatial and temporal position within the 3D tissue during the addition and removal of DMSO. Each individual cell and nucleus (including binuclear cells) was assigned a "TrackID" after segmentation. These TrackIDs

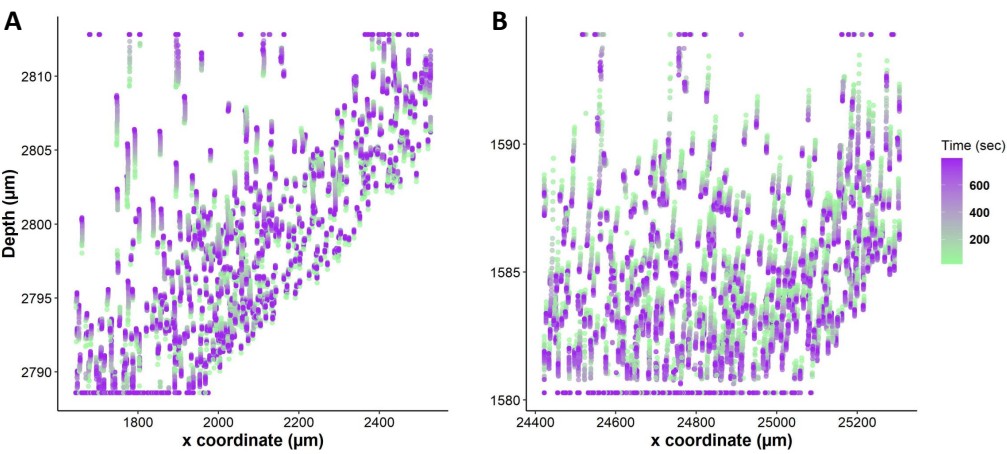

**Figure 5 Nuclei tracking within a 3D liver tissue model.** This figure illustrates nuclei tracking within a 3D liver tissue model after exposure to non-permeating anisosmotic media. Nuclei path are colored according to their positions change over time, while line (dots) length indicates how far nuclei have drifted from their original position. (A) Changes in nuclei position with hyposmotic media (80 mOsm) causing upward nuclear displacement ($n = 620$ in single tissue slice). (B) Changes in nuclei position with hyperosmotic solution (DMSO 1 mol/L) causing downward nuclear displacement ($n = 565$).

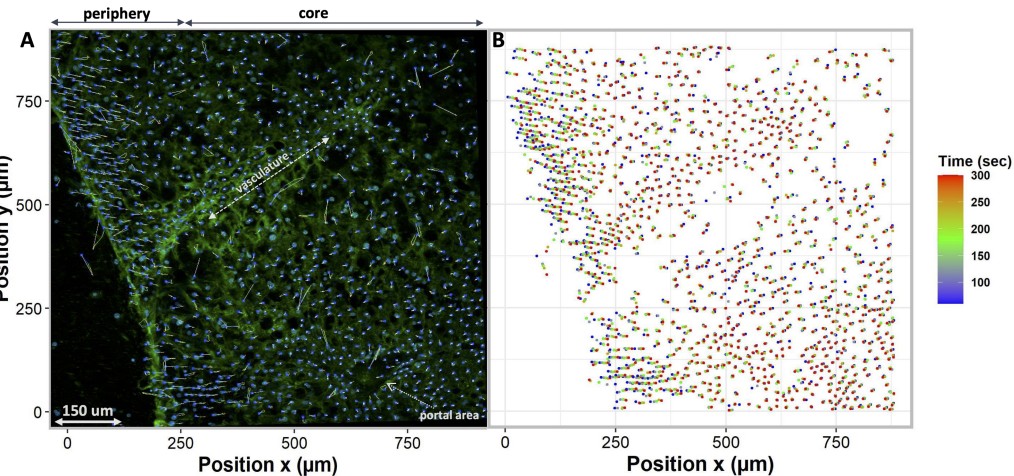

**Figure 6 Nuclei tracking in a 3D tissue model is presented in the *x* and *y* (2D) directions upon exposure to permeating DMSO (1 mol/L).** (A) Using Imaris, nuclei tracks were constructed by tracking the position of the nuclei and displayed on top of the original image. Nuclei paths are colored according to their positions and change over time, while line length indicates how far nuclei have drifted from their original position. Nuclei drift is presented within an original image of the tissue with empty spaces showing a vasculature and portal vein. (B) The change in nuclear position over time is plotted. It indicates a diffusion-limited response of nuclei, with more nuclear drift at the periphery (boundary) of the tissue as compared to the core. This is a demonstration of time dependence of osmotic stress acting on the nuclei of the tissues in a thick tissue section comprising a variety of cells.

allow the visualization each cell's *x*, *y*, and *z* positions with respect to neighboring cells as a function of time. The 3-dimensional images of the segmented cells were used to quantify the cell volume, surface area, and centroid position for these cells.

During equilibration with media containing only non-permeating solutes, cells were allowed to equilibrate with hypo and hyperosmotic media (Fig. 7). For equilibration, the changes in cell volume were continuously recorded over a 4 to 8 min period. Cells, for example, start swelling upon the addition of 80 mOsm media due to water influx as well as lower extracellular solute concentration compared to intracellular isosmotic concentration (Fig. 7A). After swelling, cells begin to equilibrate with hyposmotic media and return to their equilibrium isosmotic volume. Conversely, when 1215 mOsm media is added, cells lose water due to the higher extracellular solute concentration, resulting in a decrease in cell volume (Fig. 7B). Furthermore, it is important to note that cells swell slowly and reach equilibrium after 300 s, as opposed to hyperosmotic media which causes cells to equilibrate relatively faster at approximately 150 s. Finally, during permeating solute equilibration in liver tissue, precision cut liver slices were allowed to equilibrate with 30% DMSO (v/v). During the transition from an isosmotic solution to a hyperosmotic CPA solution, cells experienced a shrink-swell behavior as shown in (Fig. 7C).

## DISCUSSION

Monitoring cellular processes across different levels of complexity, from the cellular to the tissue scale, is a major challenge for understanding mammalian tissue structure and function. The primary challenge is to monitor these structures and their dynamic interactions non-invasively (*Megason & Fraser, 2007*). Quantification of cellular processes can help to build quantitative functional models that predict system behavior under perturbations (*Sbalzarini, 2013*). However, these models require accurate representations of the cells as well as their subcellular components in three-dimensions (3D). With the advances in microscopy, multiple fluorescent markers can be used to visualize, track and quantify these cellular processes. There are, however, certain limitations in terms of markers, tissue volume to reconstruct, measurements, high throughput computational methods, and imaging techniques. Therefore, new imaging approaches are needed for dense and thick tissues.

This study was designed to monitor the effect of permeating solute equilibration-induced osmotic stress at both the cellular and tissue scales. We presented a method for four-dimensional multichannel imaging of excised tissue slices of mouse liver. A quantitative assessment of morphological changes in living liver tissue slices was achieved with high resolution imaging provided by confocal microscopy. Several techniques are employed in this study, including fluorophore staining, confocal imaging, optical sectioning, automated segmentation, 3D reconstruction, and surface rendering. These techniques were combined to investigate morphological properties of the liver tissue undergoing osmotic stress. We quantified the effects of osmotic stress on the liver tissue by measuring three metrics of tissue morphology; first, we estimated the changes in the whole PCLS volume (Fig. 3), second, we quantified the changes in nuclei position (Fig. 5), and third, we calculated the changes in the volumetric responses of tissue embedded cells (Fig. 7).

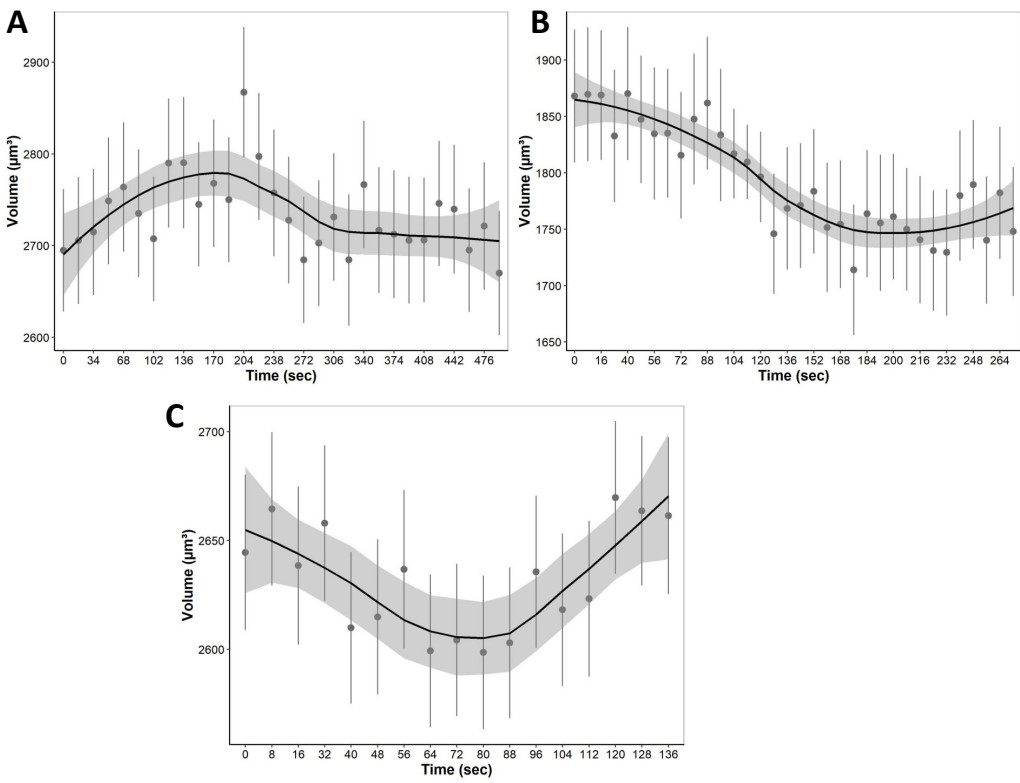

**Figure 7** **The mean volumetric responses of individual cells within 3D liver tissue after exposure to anisosmotic media.** (A) Hyposmotic media (80 mOsm) results in an increase in cell volume followed by a return towards initial volume. (B) Hyperosmotic media (1,215 mOsm) results in a decrease in volume followed by a return towards initial volume. (C) Media containing 30% DMSO (v/v) results in cells that first shrink due to the osmotic efflux of intracellular water, and then enlarge when CPA permeates, and the water is reabsorbed. The dots and error bars represent mean ± standard error of the mean measured from multiple different tissue slices. The black curved line represents the geom smoothing function, a local regression fitting method, applied to add a regression line to the scatter plot. This method is employed to add a regression line that captures the underlying trend in the data. The shaded gray region represents the 95% confidence interval for the fitted values. At least 530 cells were used for each data point.

This study presented a novel imaging approach to measure dense and thick tissues, allowing cells and nuclei to be measured within a three-dimensional structure, visualizing, tracking, and quantifying osmotic stresses, and evaluating the volumetric response of cells in response to various osmotic conditions. Furthermore, our research suggests that cells change their volume in response to osmotic stress, but the response is diffusion limited and depends on CPA concentrations, exposure time, and tissue thickness. In addition, our results indicate that nuclei undergo directional displacement when exposed to osmotic stress, and the degree of displacement depends on the direction and amount of osmotic flux (Figs. 5 and 6). These multiscale assessments and quantifications of osmotic stresses will provide insights into liver cell biology, tissue physiology, and mechanobiology.

It is well known that cells and tissues undergo osmotic stresses during cryopreservation (*Fahy et al., 2013*; *Kaur, Pramanik & Sarangi, 2013*; *Benson & Macklin, 2015*), however, it is

not clear how cells within a complex, multicellular, heterogeneous tissue respond to these mechanical stresses. There has been some recent success in sub-zero preservation of tissues and organs by leveraging machine perfusion technology for preconditioning and unloading of high osmolality CPAs (*Tessier et al., 2022*). However, these preservation processes are still associated with mechanical damage (osmotic stress) due to differences in solute concentration across membranes and within interstitium. Therefore, the quantification of mechanical stresses can assist in the design of an optimized method for loading and unloading CPAs with high osmolarities.

Our study reveals that osmotic stresses in tissues can cause nuclear displacement (Figs. 5 and 6). This study represents the first visualization and quantification of intracellular osmotic stresses in liver tissue by using nuclei as a universal probe. In view of the importance of nuclear position in determining the function and fate of a cell (*Fedorchak, Kaminski & Lammerding, 2014*), we believe that it is important to understand the effects of osmotic stress acting on nuclei in liver tissue. We speculate that by accurate tracking of nuclei within a three-dimensional tissue, we can accurately estimate how much osmotic forces (stress) are acting on that spatio-temporal region of the tissue, as earlier described by *Khavari & Ehrlicher (2019)* who quantitatively assessed nuclei deformation to reveal pressure distribution in a 3D cell cluster tissue. Another study presented a complex algorithm for reconstructing and analyzing 3D tissue architecture at multiple scales in liver tissue (*Sbalzarini, 2013*). Altogether, studies have shown that nuclei can be used as an ideal probe for compressive forces and can be employed to determine the effects of external osmolarity in thick three-dimensional tissue (*Khavari & Ehrlicher, 2019*). Therefore, we build on this previous work and offer the use of nuclei as a probe for mechanical stress and strain.

In this study, we aim to quantify the effects of osmotic stress at the tissue level by measuring the displacement of nuclei caused by osmotic stress. By accurately tracking nuclei within a three-dimensional tissue, we may be able to predict the magnitude of mechanical forces (stress) acting on that spatiotemporal region of the tissue. Figure 5 illustrates the tracking of nuclei within a 3D liver tissue model following exposure to nonpermeating anisosmotic media, highlighting the observed upward and downward nuclear displacement. The observed changes in nuclear orientation are correlated with the unique polarity of liver cells, as elucidated in previous studies (*Müsch, 2014*). The observed displacement may signify cellular-level damage within the tissue, as deformation of the nucleus is a well-documented response to various extracellular sources of stress. Such stressors have been linked to increased nuclear rupture, alterations in gene expression, and the accumulation of DNA damage (*Shah et al., 2021*). Additionally, nuclear abnormalities associated with biologic disorders, including those characterizing malignancies and cancer, have been extensively discussed in the literature (*Denais & Lammerding, 2014*). Moreover, the technique used in this study can be useful for examining the movement and interaction of different cell types in their native environment. A quantitative analysis of the tissue is possible at various depths within the tissue using this technique.

A variety of studies have previously used this multispectral imaging technique to study tissue biology. For instance, *Marques et al. (2015)* used intravital microscopy for imaging liver tissue biology *in vivo* using confocal microscopy. They have used antibodies to stain

different cells in liver tissue, such as hepatocytes, endothelial cells, and leukocytes under physiological or pathological conditions. However, our research involves the investigation of cell and nuclear changes within a 3D liver tissue model by adding and removing various cryoprotectant solutions. Moreover, another study by *Johnson & Rabinovitch (2012)*, employed fluorescent dyes to image excised liver tissue, also used confocal microscopy imaging to assess tissue viability. However, we were able to expand the technique well beyond their previously published 100 μm thick tissue. Similarly, others have used this approach to study cell and molecular dynamics during chick embryogenesis (*Teddy, Lansford & Kulesa, 2005*), bile canalicular and sinusoidal microarchitecture in the liver (*Hammad et al., 2014*), epithelial tissue inflammation responses in zebrafish (*Enyedi et al., 2013*), ex-vivo tissue labeling of human and mouse prostate tissue (*Burgstaller et al., 2015*), and human lung tissue slices (*Burgstaller et al., 2015*; *Blatter, 1999*). Our work involves the addition and removal of different permeating or non-permeating cryoprotectants solutions to investigate the dynamics of cell and nuclear changes within a 3D liver tissue model.

In this study, we used a workflow of microscopic imaging and image analysis to estimate the changes in cell volume within heterogeneous liver tissue. Our results (Fig. 7) suggest that cells are undergoing volume regulation in response to changes in extracellular solute concentration. As liver tissue contains a variety of cells, including hepatocytes, Kupffer cells, stellate cells, and endothelial cells (*Ishibashi et al., 2009*), this variation in cell size and volume may explain the large volume variance observed in Fig. 7. By accurately estimating cell volume changes and tracking them in three dimensions, we can estimate the solute concentration as well as the transport kinetics of water and solutes across membranes within the tissue as described by *Warner et al. (2021)* for articular cartilage tissue. However, more complex mass transfer modeling is required to accurately model these individual cell volume kinetics and will be done in future studies. Similarly, these cell and nuclei volume and tracking data can be utilized to fit agent-based models to liver tissue to simulate these conditions as shown by (*Wang, Heiland & Macklin, 2019*) for liver tissue mechanobiology. After a model is constructed, it can be used to simulate mechanical damage caused by osmotic pressure during equilibration with cryoprotectant media. Moreover, this technique provides first-of-its-kind insight into the dynamics of cell volume, cell to cell interaction in a multicellular structure in their native environment.

Under significant osmotic challenges, cell volume regulation plays a vital role in maintaining cell homeostasis (*Corasanti, Gleeson & Boyer, 1990*; *Sardini et al., 2003*). Cells initiate this regulatory mechanism when they swell or shrink in an anisosmotic medium. Accordingly, cells need to alter the activity of specific ion channels and transporters during anisosmotic stress to return to isosmotic volume. For instance, when a cell becomes osmotically swollen, the process of regulatory volume decrease (RVD) causes it to release KCl, osmolytes, and water, which reduces its volume towards its original (*Hoffmann, Lambert & Pedersen, 2009*). Conversely, osmotically shrunken cells gain KCl and water, increasing their volume to their original value, a process known as regulatory volume increase (RVI) (*Wehner & Tinel, 1998*; *Okada, 2004*). Moreover, RVI has been coupled with the stimulation of $Na^+$-$K^+$ pumps and ($Na^+$-$K^+$-$2Cl$) cotransporters (NKCC) in many cell types (*Hoffmann, Lambert & Pedersen, 2009*). Similar volume regulation is

displayed for cells within a tissue in Fig. 7, which indicates that cells are regulating their volume in response to osmotic stress.

There are at least two potential limitations concerning the results of this study. The first limitation concerns the oxygenation of tissues during imaging. Liver tissue is one of the most important metabolizing tissues in the body, containing the highest concentration of mitochondria, with each hepatocyte containing 1,000 to 2,000 mitochondria (*An et al., 2020*). Therefore, the viability of liver tissue depends on a considerable amount of oxygen. It is possible, however, that liver tissue isolation, preparation of precision cut tissue slices, and imaging, until the CPA has been fully equilibrated, may disturb the oxygen availability within hepatic lobules. Therefore, it may have an adverse effect on the tissue's long-term viability. Consequently, having access to a microscope with an incubation chamber set at a lower temperature and continuous oxygenation during the acquisition of the image would be ideal, since imaging also contributes to the warming of the tissue (*Happel, Thatenhorst & Dietzel, 2012*). A second potential limitation is the use of a microfluidic device with a chamber for holding tissue and allowing directional fluid flow. A directional flow of fluid is essential for estimating fluid fluxes and mechanical properties of tissues and predicting mass transfer from extracellular space to cells (*Warner et al., 2021*; *Jiang & Sun, 2013*).

## CONCLUSION

In summary, our research describes the effects of osmotic stresses on cell volume and nuclear displacement within a tissue, however, a further investigation of the biological effects of these stresses needs to be done. Our research stimulates further investigation of questions such as whether these nuclear displacements are transient and reversible, how cell volume perturbations can trigger cell signaling or damage pathways, and whether these two factors influence long-term tissue viability or functional outcome. Therefore, future work will involve measurement of the biological consequences of the volumetric changes and nuclear displacements, as well as the effects of osmotic stress on the long-term viability of liver tissues. Finally, the methods described in this study can also be adapted to examine other cryobiologically relevant tissues, including those of the ovary, kidney, heart, pancreas, and brain.

### Funding

This work was funded by the National Science and Engineering Research Council (RGPIN-2023-04007) and the Canadian Institutes of Health Research (PJT–175283). The funders had no role in study design, data collection and analysis, decision to publish, or preparation of the manuscript.

### Grant Disclosures

The following grant information was disclosed by the authors:
National Science and Engineering Research Council: RGPIN-2023-04007.
Canadian Institutes of Health Research: PJT–175283.

## Competing Interests

The authors declare there are no competing interests.

## Author Contributions

- Iqra Azam conceived and designed the experiments, performed the experiments, analyzed the data, prepared figures and/or tables, authored or reviewed drafts of the article, and approved the final draft.
- James D. Benson conceived and designed the experiments, authored or reviewed drafts of the article, and approved the final draft.

## Animal Ethics

The following information was supplied relating to ethical approvals (i.e., approving body and any reference numbers):

University of Saskatchewan Animal Care Committee

## Data Availability

The data is available at the Federated Research Data Repository: James, B., Iqra, A. (2023). Multiscale transport and 4D time-lapse imaging in Precision Cut Liver Slices (PCLS)-Supplementary Data. Federated Research Data Repository. Available at https://doi.org/10.20383/103.0800.

## Supplemental Information

Supplemental information for this article can be found online at http://dx.doi.org/10.7717/peerj.16994#supplemental-information.

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
