# Peer review of "Multiscale transport and 4D time-lapse imaging in precision-cut liver slices (PCLS)"

_PeerJ, doi:10.7717/peerj.16994_

## Round 0.1 · original submission · Minor Revisions

Both reviewers' comments are reasonable and constructive. I hope your paper will be revised according to their comments.

Reviewer 1 ·

Basic reporting

Overall, the paper is well written. However, I have some minor suggestions for improvement.

Line 16: three dimensional is already defined as 3D above, so you can just use 3D here

Line 19: Do you mean “metrics” or “metrices”?

The introduction is quite long and does not seem very focused. It would be helpful to more clearly present how each paragraph in the introduction relates to the hypothesis/problem being addressed in this paper. For example, it is not clear how it is useful to the reader to know that “nuclei are the largest and stiffest organelle” (line 131).

Line 60-61: it is not clear that it is necessary for damage models to account for all of these factors, and it is also not clear that the technique presented in this paper can measure all of them.

Line 305: there is a period missing after “segmentation”

Line 330: “in three dimensions” instead of “in a three-dimensions (3D)”

Experimental design

This work addresses the need for experimental methods to track dynamic changes in tissues across multiple scales, from subcellular to the entire tissue slice. The objective is clearly presented. Overall, the methods are clear and sufficient detail has been provided.

However, it would be helpful to add more detail about how the tissues were “rapidly” exposed to anisosmotic solutions and the time scale for the change in solution composition. This is relevant to interpretation of the results.

More information should also be provided about how the curves were created in Figure 7. What is the “geom” smoothing function?

Validity of the findings

Overall, the main conclusions are supported by the data. However, some interpretations of the results may be questionable (see below)

Line 277: the word “significantly” implies statistical significance. However, statistical analysis was not performed (n=1 for each condition). I’d recommend using “substantially” instead. It might be possible to statistically analyze the results using a “sign test” to test the hypothesis that the tissue volume is lower after exposure to hypertonic solution.

Figure 7: It is not clear whether any of the changes in cell volume are statistically significant. For example, in panel A all error bars appear to overlap. It might be easier to see differences if the cell volume data were normalized to the initial cell volume for each cell.

Reviewer 2 ·

Basic reporting

The topic is very interesting and relevant for the field of 3D models. I recommend to add in the introduction the clear aim of the study and the implications in experiments with PCLS to improve the clarity of the article.

Experimental design

Considering the slice variability (in terms of vasculature, and other features), it is inadequate to use n=1 for the volumetric responses (Fig 3) as this may lead to inaccurate conclusions. The authors should increase the number of slices analysed or at least discuss this limitation in the discussion.
Relevant controls with untreated slice are missing in Fig 6.

Validity of the findings

The clarity of the message would improve if the authors describe more in details the meaning of upward and downward nuclear displacement especially given that up and down orientation is completely arbitrary (Fig 5) - is it related to hepatocyte polarity? What is the biological implication? The authors could support their statements adding references on this observation.

---

## Round 0.2 · accepted · Accept

I acknowledge that the authors have addressed all reviewers' comments, and am happy with this current version. Now this manuscript is ready for publication.